# Genetic Variants Determine Treatment Response in Autoimmune Hepatitis

**DOI:** 10.3390/jpm13030540

**Published:** 2023-03-17

**Authors:** Stephan Zandanell, Lorenz Balcar, Georg Semmler, Alex Schirmer, Isabella Leitner, Lea Rosenstatter, David Niederseer, Karl Sotlar, Anna-Maria Schneider, Michael Strasser, Sophie Gensluckner, Alexandra Feldman, Christian Datz, Elmar Aigner

**Affiliations:** 1First Department of Medicine, Paracelsus Medical University, Müllner Hauptstrasse 48, 5020 Salzburg, Austria; 2Division of Gastroenterology and Hepatology, Department of Internal Medicine III, Medical University of Vienna, Währinger Gürtel 18-20, 1090 Vienna, Austria; 3Department of Internal Medicine, General Hospital Oberndorf, Teaching Hospital, Paracelsus Medical University, Paracelsusstrasse 37, 5110 Oberndorf, Austria; 4Department of Cardiology, University Hospital Zurich, Rämistrasse 100, 8091 Zurich, Switzerland; 5Institute of Pathology, Paracelsus Medical University, Müllner Hauptstrasse 48, 5020 Salzburg, Austria; 6Department of Pediatrics, Paracelsus Medical University, Müllner Hauptstrasse 48, 5020 Salzburg, Austria

**Keywords:** AIH, autoimmune liver disease, genetic polymorphism, genotyping, HSD17B13, MBOAT7, single-nucleotide polymorphisms, PNPLA3, TM6SF2, treatment response

## Abstract

Background: Autoimmune hepatitis (AIH) is a rare entity; in addition, single-nucleotide polymorphisms (SNPs) may impact its course and outcome. We investigated liver-related SNPs regarding its activity, as well as in relation to its stage and treatment response in a Central European AIH cohort. Methods: A total of 113 AIH patients (i.e., 30 male/83 female, median 57.9 years) were identified. In 81, genotyping of PNPLA3-rs738409, MBOAT7-rs626238, TM6SF2-rs58542926, and HSD17B13-rs72613567:TA, as well as both biochemical and clinical data at baseline and follow-up, were available. Results: The median time of follow-up was 2.8 years; five patients died and one underwent liver transplantation. The PNPLA3-G/G homozygosity was linked to a worse treatment response when compared to wildtype [wt] (ALT 1.7 vs. 0.6 × ULN, *p* < 0.001). The MBOAT7-C/C homozygosity was linked to non-response vs. wt and heterozygosity (*p* = 0.022). Male gender was associated with non-response (OR 14.5, *p* = 0.012) and a higher prevalence of PNPLA3 (G/G vs. C/G vs. wt 41.9/40.0/15.0% males, *p* = 0.03). The MBOAT7 wt was linked to less histological fibrosis (*p* = 0.008), while no effects for other SNPs were noted. A polygenic risk score was utilized comprising all the SNPs and correlated with the treatment response (*p* = 0.04). Conclusions: Our data suggest that genetic risk variants impact the treatment response of AIH in a gene-dosage-dependent manner. Furthermore, MBOAT7 and PNPLA3 mediated most of the observed effects, the latter explaining, in part, the predisposition of male subjects to worse treatment responses.

## 1. Introduction

Autoimmune hepatitis (AIH) is a clinical entity known for its heterogeneity at presentation, its varying disease course, and its different long-term outcomes [1,2,3]. Up to 85 percent of patients respond to steroids and first-line immunosuppression with azathioprine, thereby achieving fast and durable clinical remission [4,5]. Still, only 10–15% reach partial remission with continuously elevated transaminases. In addition, a minor proportion even progress to end-stage liver disease despite therapy, thus requiring liver transplantation, or instead dying from cirrhosis-related complications [6,7,8,9].

Underlying factors contributing to either favorable or eventually fatal outcomes have been thoroughly investigated. Only a few features were found to be potential early markers of disease progression: a young age at presentation, an acute severe onset, slow treatment response, and cirrhosis at baseline [2,9,10,11]. Male gender was identified as a risk factor for adverse outcomes including development of hepatocellular carcinoma (HCC) in AIH [2]. Regarding genetic variants, the HLA-DRB1*0301 and HLA-DRB1*0401 genotypes appear to be associated with disease development and outcomes in Northern European Caucasians [9,12,13,14]. In a comparison of Italian and North American cohorts, the HLA-B8-DR3-DQ2 and HLA-DR4 haplotypes occurred more frequently in Northern American AIH patients than in their Italian counterparts [15]. Concerning other haplotypes, Italian patients with hepatitis C were more likely to show HLA-DR7, thus predisposing them to a formation of liver kidney microsome (LKM) antibodies [16]. No particular autoantibody profile has been conclusively linked to a disease course [2,9,17,18]. However, estimating an individual patient’s risk for progressive disease is mostly a matter of conducting a clinical observation on the response to immunosuppressive treatment. In 2020, a German transplant center published the first data on the PNPLA-rs738409 G/G genotype, which is associated with a higher risk for liver transplantation or death, as well as higher non-invasive fibrosis scores in AIH patients [19]. Of the overall 239 patients, 4/12 (33.3%) with the homozygous PNPLA3 variant underwent liver transplantation, or died during follow-up. Other single-nucleotide polymorphisms (SNPs), which have been related to liver disease as mutations of TM6SF2-rs58542926 and MBOAT7-rs641738, or through the presence of HSD17B13-rs72613567:TA insertion, were not associated with either detrimental or beneficial effects on the disease course. Another Polish study from 2021, which included 313 AIH patients, could not confirm the effect of the PNPLA3 polymorphism on the clinical endpoints in AIH, but demonstrated a link of the PNPLA3 risk allele to higher MELD scores. Only a carriage of the protective MARC1 variant was found to be associated with lower AST, ALT, and APRI scores [20]. In general, genetic variants were associated with progression among both rare and common liver diseases, including both alcoholic and non-alcoholic fatty liver diseases, as well as hemochromatosis, hepatitis C, and Wilson’s disease [21,22,23,24,25,26,27,28,29,30]. Hence, investigation into the role of these genetic variants that are linked to various liver diseases is also helpful in terms of potentially improving risk assessments regarding subjects with AIH.

We, therefore, aimed to analyze the clinical outcome of all 113 AIH patients from our liver disease database with regard to their genetic risk profile.

## 2. Methods

### 2.1. Patients

Our study was based on a retrospective chart review. We identified 113 cases of AIH between 1996 and 2021 at the time of first referral to the liver outpatient clinics of Paracelsus Medical University Hospital, Salzburg, Austria, and Hospital Oberndorf, Austria. In the majority of patients (97/113, 85.8%), the diagnosis of AIH was established at or shortly after first referral in accordance with the simplified diagnostic criteria of the International Autoimmune Hepatitis Group (IAIHG), including autoantibody positivity, elevated IgG, compatible liver histology, and the absence of viral hepatitis [31]. Liver histology that was compatible with AIH was available in 97/113 (85.8%) of patients. For patients that were diagnosed before 2008, the revised original scoring system for AIH [32] from 1999 was applied. In addition, the latter subjects that were included in the analysis were diagnosed as AIH on the basis of their liver biopsy results and treatment responses.

We were able to determine the clinical course and the genotyping of liver-disease-related SNPs in 81 patients (71.7%). For each patient, a detailed workup, including past medical history, anthropometric parameters, and laboratory evaluation were available. A liver stiffness measurement (LSM) using Fibroscan (Echosens, Paris, France) was documented when available. The response to immunosuppressive treatment was assessed by clinical and biochemical re-evaluation after the initiation of immunosuppression at intervals of 1, 3, 6, 12, and up to 36 months. The classification of the patients’ treatment response was based on the recently published consensus statement of the International Autoimmune Hepatitis Group (IAIHG) [33]. However, these recommendations have not yet been implemented in the AIH guidelines of the European (EASL) or American liver associations (AASLD) [34,35]. The clinical endpoints were mortality and end-stage liver disease (ESLD), comprising liver cirrhosis, transplantation, and the occurrence of HCC. Cirrhosis was defined as the following histological or clinical features being present: (a) liver histology displaying cirrhotic changes, i.e., fibrosis stage 4 and higher; and/or (b) clinical signs of both compensated or decompensated cirrhosis, defined as ascites, encephalopathy, or variceal bleeding. The treatment response was assessed by an evaluation of transaminases and immunoglobulin G (IgG) at predefined time points according to the 2022 IAIHG consensus statement [33]. Four groups were defined: (1) complete response (CR); (2) insufficient response (IR); (3) non-response (NR); and (4) no therapy (NT). The criteria for CR were the ‘normalization of serum transaminases and IgG below the upper limit of normal (ULN) up to 6 months after initiation of treatment’. IR was established as the failure to meet the CR criteria. NR was defined as a ‘<50% decrease of serum transaminases within 4 weeks after initiation of treatment’. Patients in the NT group received no immunosuppressive therapy and were therefore excluded from the calculations regarding the comparison of treatment response groups.

The present study was approved by the Ethics Commission of Salzburg (application no. 1026/2021).

### 2.2. Laboratory Evaluation and SNP Genotyping

At baseline and subsequent follow-up (FU) visits, biochemical characterization comprised the following tests: full blood count; electrolytes; kidney and liver function tests; serum iron parameters; C-reactive protein; fasting glucose; lipid profile; hepatitis A, B, and C serology and polymerase chain reaction (PCR); copper; ceruloplasmin; prothrombin time; serum protein electrophoresis; and the quantification of immunoglobulins. The baseline panel also included autoantibody screening for AMA, ANA, APCA, ASMA, LC-1, LKM, SLA, and Aktin antibodies. All parameters were measured by standardized automated laboratory methods after an overnight fasting period.

The SNP genotyping was conducted first by the extraction of the genomic DNA of EDTA-treated whole blood samples. Then, PCR analyses using TaqMan SNP probes from Applied Biosystems (Foster City, CA, USA) were conducted according to the manufacturer’s protocols on the ViiA7 instrument (Applied Biosystems, Forster City, CA, USA). By means of allelic discrimination, the following SNPs were investigated: PNPLA3 rs738409 C>G, TM6SF2 rs58542926 C>T, MBOAT7 rs626238 G>C, and HSD17B13 rs72613567:TA. The genotyping for HFE and SERPINA1 (alpha-1-antitrypsin) has been performed routinely since 1996 and other SNPs were implemented continuously as part of routine liver disease diagnostic work-up as they emerged in the literature at our centers. In our outpatient department, we routinely collect serum specimens of new patients and store them at −20 °C for future genetic analyses. Written consent for this procedure is obtained from every patient. Missing SNPs were determined from those stored DNA samples.

All patients were categorized according to the presence and frequency of the variant allele. Therefore, four separate analyses were conducted (wildtype [wt] vs. heterozygous [het] vs. homozygous [hom] variant allele carriers): PNPLA3-rs738409 C/C vs. C/G vs. G/G genotype, MBOAT7-rs626238 G/G vs. G/C and C/C, TM6SF2-rs58542926 E/E vs. E/K (K/K homozygosity not detected), and HSD17B13-rs72613567 A/A vs. A/dupA vs. dupA/dupA genotype.

### 2.3. Liver Biopsy Workup

Biopsies were examined by pathologists of our local University Institute of Pathology. The grading (disease activity) and staging (fibrosis stage), as per Batts and Ludwig, were performed for each patient. Liver tissue samples were routinely stained with hematoxylin and eosin, as well as with a Masson-trichrome stain for connective tissue and Perl stain for iron. The main histological features documented for each biopsy included interface hepatitis, portal inflammation, centrilobular necrosis, bridging fibrosis, ductular proliferation, and bile duct damage. These features were reported as either present or absent. The pathology results were analyzed regarding the absence or presence of fibrosis, the presence of cirrhosis, as well as the presence and degree of steatosis. Steatosis grades were defined as follows: stage 0 when no steatosis or when <5% of the hepatocytes were affected; stage 1 for 5–33%; stage 2 for 33–66%; and stage 3 when >66% of the hepatocytes showed fat accumulation.

### 2.4. Statistical Analysis

Statistical analyses were performed using SPSS (SPSS 29.0, IBM Statistics, Armonk, NY, USA). Variables are presented as the median (interquartile range [IQR] in brackets). The distribution of datasets was assessed via a Kolmogorov Smirnov test. Comparisons between groups were calculated by using the Chi-square test, Mann–Whitney U test, the Kruskal–Wallis H test—with the Bonferroni correction—and two-way mixed ANOVA, as appropriate.

The influence of genetic risk variants was evaluated by a comparison of wt versus het versus the hom carriers of the risk alleles. The observed allele frequency in our study was tested for conformation to an ideal population in a Hardy–Weinberg equilibrium by calculating the Chi-square values.

To evaluate a potential combined effect of the multiple SNPs on the treatment response, we calculated a polygenic risk score for each patient and performed a linear regression comprising the individual polygenic risk score and assigned treatment response group.

To identify factors associated with non-response, a binomial regression analysis with the following covariates was calculated: age, gender, PNPLA3, TM6SF2, MBOAT7, and HSD17B13 status.

A two-sided *p*-value < 0.05 was considered significant.

## 3. Results

### 3.1. Patient Cohort Baseline Characteristics

An overview of the study cohort is given in Figure 1. Of the 113 patients diagnosed with AIH, the genotyping and parameters for assessment of the clinical course were available in 81, and these subjects comprised the follow-up cohort. Twenty-six patients had no genotyping results available and in six no follow-up visits had been recorded.

Female patients were predominant in the whole cohort (*n* = 82; 72.6% of 113 patients). The median age at diagnosis was 57.9 years (IQR 46.6–67.8), and the median BMI was 25.0 kg/m^2^ (IQR 22.0–27.5). Further, 14 of the 113 patients (12.4%) showed histological or clinical signs of liver cirrhosis at baseline (BL). The characteristics of the BL cohort are summarized in Table 1.

### 3.2. Prevalence of Risk Alleles

All SNPs of interest were analyzed in comparison to an ideal population in the Hardy–Weinberg equilibrium. Each single-risk variant allele (PNPLA3-rs738409 G, TM6SF2 rs58542926 K, MBOAT7 rs626238 C, and HSD17B13 rs72613567 dupA) was as prevalent as expected. Details are given in Appendix A. No associations between the four SNPs were noted via the calculation of the distribution of PNPLA3, TM6SF2, MBOAT7, and HSD17B13, according to the respective other variants.

### 3.3. Activity and Stage of Liver Disease at Diagnosis

Analyses of the biopsy specimen for the activity and stage of AIH at diagnosis revealed effects only for the MBOAT7 subgroups. The het MBOAT7 carriers had the lowest numbers of activity score A1 (het 4/33 [12.1%] vs. wt 9/21 [42.9%] vs. hom 7/16 [43.8%], *p* = 0.016). Activity A2 was least likely to be found in wt MBOAT7 patients (wt 2/21 [9.5%] vs. het 14/33 [42.4%] vs. hom 4/16 [25.0%], *p* = 0.031). No differences were noted for A3 activity. Regarding stage, MBOAT7 wt was linked to less fibrosis vs. het and hom risk allele carriers in the biopsy specimen at BL (Fibrosis grade 0: wt 45.5% vs. het 9.1% vs. hom 23.5%; fibrosis grade 1–3: wt 45.5% vs. het 84.8% vs. hom 70.6%; both *p* = 0.008).

### 3.4. Response to Therapy

Evaluation of the biochemical response to immunosuppression (Figure 2) at the predefined FU time points showed a faster response for wt PNPLA3 patients, as well as significantly lower ALT levels when compared to both het and hom risk variant carriers at the last FU, which was 32.5 months after BL (mean ± SD: wt 0.6 ± 0.2 vs. het 1.1 ± 0.9 × ULN, *p* = 0.042; wt vs. hom 1.7 ± 0.4 × ULN, *p* < 0.001). However, when comparing the ALT course of all three groups over the whole time range from BL to the last FU, only a trend for a better response in wt carriers was detected by a calculation of a two-way mixed ANOVA. The assumption of sphericity was violated because of the high differences in variances (*p* < 0.001), such that the corrected *p*-value after Huynh-Feldt was used (wt vs. het, *p* = 0.229; wt vs. hom, *p* = 0.143). No differences in ALT courses were noted between the TM6SF2, HSD17B13, and MBOAT7 subgroups.

The assessment of treatment response according to the IAIHG guidelines (CR, IR, NR) revealed no differences between wt and risk allele carriers of HSD17B13, PNPLA3 and TM6SF2 subgroups. Hom MBOAT7 patients were more likely to be non-responders compared to het and wt MBOAT7 patients (wt 0/23 vs. het 3/36 [8.3%] vs. hom 4/15 [26.7%], *p* = 0.022). None of the four SNPs was more frequent in the complete or partial remission subgroups. For further details, see Appendix A.

A comparison of the anthropometric and biochemical characteristics at BL for all SNP subgroups yielded only minor differences. Male gender was more prevalent in the het and hom carriers of the PNPLA3 risk variant than in the wt group (wt 15.0% vs. het 41.9% vs. hom 40.0%, *p* = 0.03). The hom PNPLA3 patients tended to be younger at the presentation age (wt 55.3 vs. het 58.0 vs. hom 45.6 years, *p* = 0.072), although this did not reach the level of significance. For HSD17B13, both het and hom mutant allele carriers had a higher prevalence of diabetes. Regarding TM6SF2, the carriage of the risk allele was associated with higher platelets and a higher prevalence of grade 2 steatosis than TM6SF2 wt. Considering MBOAT7, the wt patients had a lower WBC, and also lower hemoglobin at BL (for details, see Appendix A).

Due to the low numbers of the predefined endpoints in our cohort (mortality *n* = 5, ESLD *n* = 16), no differences in the clinical endpoints could be detected (Table 2 for details).

### 3.5. Elderly Patients

To evaluate a possible effect of age on the occurrence of variant alleles and treatment responses, we compared younger (<65 years) and elderly (≥65 years) AIH patients, similar to the study of Granito et al. [36]. Our cohort comprised a similar number of elderly patients ≥ 65 years (36/114, 32%) when compared to the mentioned Italian AIH cohort (20/76, 26%). The distribution of HSD17B13, TM6SF2, and MBOAT7 variant alleles between the younger and elderly patients were not different. Only the heterozygous PNPLA3 variant was more frequent among the elderly vs. younger patients (14/23 [60.9%] vs. 17/58 [29.3%], *p* = 0.008). The elderly were similar to the younger AIH patients regarding treatment response.

### 3.6. Polygenic Risk Score

As described earlier by our group [37], we established a polygenic risk score [PRS], which was calculated for each patient based on the individual genetic risk profile. Each copy of a risk allele of MBOAT7, PNPLA3, and TM6SF2 was assigned 1 point—i.e., heterozygosity added 1 and homozygosity added 2 points to the score. Copies of HSD17B13, on the other hand, reduced the risk score by either one (heterozygosity) or two (homozygosity) points. Therefore, the possible overall score ranged from a minimum of −2 up to +6. An overview of the distribution of the PRS values is given in Table 3.

A calculation of the mean PRS per treatment response group revealed a linear, direct proportional change from the lower values of complete remission to the higher scores for the non-response group. A linear regression confirmed that average PRS could statistically significantly predict the treatment response group, i.e., *F*(1.72) = 4.39, *p* = 0.04. The regression equation was as per the following: the predicted treatment response group = 1.547 + 0.112 × PRS.

### 3.7. Risk Factor Analysis for Non-Response

We conducted a binomial logistic regression to identify risk factors associated with non-responses compared to both complete and partial responses. Age, gender, and presence of the HSD17B13 variant, as well as the presence of the MBOAT7, PNPLA3, and TM6SF2 risk alleles were used as the covariates. Due to multicollinearity, MBOAT7 had to be removed from the regression model. Only male gender (odds ratio 14.455 [95% CI 1.806–115.672], *p* = 0.012) showed a significant impact on the likelihood for non-response (Table 4 for details).

## 4. Discussion

Data regarding the clinical course of liver diseases in relation to genetic risk profiles apart from NAFLD is scarce, especially in the field of autoimmune liver diseases. Prompted by the clinical observation in daily practice that some male AIH patients harboring the PNPLA3 risk allele presented with adverse clinical outcomes and slow treatment response, we conducted a detailed investigation of all our AIH patients.

In the present study, we found the following: (1) Lower histological fibrosis stages in MBOAT7 wt patients; (2) higher ALT levels in het and hom PNPLA3 risk allele carriers when compared to wt patients in the long-term FU; (3) a higher prevalence of male gender in the PNPLA3 risk variant groups; (4) a higher rate of non-responders in hom MBOAT7 risk allele carriers; (5) a linear correlation between polygenic risk scores and impaired treatment responses; (6) no difference in clinical endpoints (death or ESLD) for all SNP subgroups; and (7) no impact of HSD17B13 and TM6SF2 on the AIH disease course nor outcome.

In 2020, Mederacke et al. provided evidence that the carriage of the PNPLA3-rs738409 risk allele in AIH patients is associated with a worse overall outcome, including a higher mortality and a higher need for transplantation [19]. No other liver-related SNP was associated with clinical outcomes in this study. Our study could not attribute outcomes, but could find a link of ALT courses and overall treatment responses to different SNPs.

With regard to baseline characteristics, our data revealed a potentially “protective” effect of MBOAT7 wt on the stage of fibrosis in AIH as a novel finding. Different fibrosis stages of AIH on the index biopsy have not yet been attributed to a single SNP. Mederacke et al., in 2020, reported higher values of non-invasive fibrosis scores for the PNPLA3 risk allele carriage [19]. However, biopsy samples in this study did not confirm the difference in fibrosis staging for carriers of PNPLA3 variant alleles. An increased risk of liver fibrosis associated with the MBOAT7 polymorphism has been shown for other chronic liver diseases, such as hepatitis B, hepatitis C, and NAFLD [38,39,40]. With regard to this broad spectrum of etiologies it appears reasonable that a higher risk of fibrosis could also be attributed to MBOAT7 down-regulation in AIH, thereby suggesting a common underlying mechanism in chronic liver diseases.

ALT is universally available and the most important biochemical marker in terms of monitoring the course of AIH. Higher levels are associated with higher disease activity. The normalization of ALT is the goal of any AIH-directed therapy; as such, higher levels in PNPLA3 mutant allele carriers after 2.8 years serve as a surrogate marker for a worse treatment response.

The “gender gap” of AIH, meaning a higher prevalence in women vs. men, is a well-known fact that is similar to many other autoimmune diseases [41,42]. Considering the sex-specific outcomes in AIH, the available literature has reported a higher risk for unfavorable outcomes in males. Two studies from Denmark and Japan found male gender to be associated with worse outcomes in terms of higher mortality or a diminished response to therapy, respectively [2,43]. Al-Chalabi et al. described a younger age of disease onset combined with a higher relapse rate in 51 male vs. 187 female AIH patients [44]. The authors interpreted these observations as a possible effect of a higher prevalence of the HLA-A1-B8-DR3 haplotype in men, which is associated with a higher susceptibility to AIH, a younger presentation age, and a higher rate of relapses [45]. A 2002 study from Czaja et al. reported a similar outcome of 34 men vs. 134 women treated for AIH at the Mayo Clinic in Rochester, USA [46]. Treatment failure was more common only in HLA-DR3 positive men when compared to HLA-DR4 women. The study from Mederacke et al. found liver transplantation and death to be associated with the PNPLA3 G/G genotype, non-remission under therapy, and cirrhotic changes in the index biopsy, but not in regard to the male gender [19].

Our study identified a higher prevalence of the male gender in both PNPLA3 risk allele groups. We, therefore, suggest that male AIH patients with both het or hom PNPLA3 risk variant carriages may be at a higher risk for an adverse disease course. Male gender itself was the only factor associated with an increased likelihood of non-response (OR 14.5, *p* = 0.012) in our cohort. As PNPLA3 risk alleles were higher in males, PNPLA3 alleles can potentially explain part of the “gender gap” observations in AIH.

A comparison of age-related frequencies of SNPs between younger and elderly (≥65 years) AIH patients did not yield relevant differences, except with respect to PNPLA3 heterozygosity, which was found more often in elderly patients. However, this minor difference did not have an impact on treatment response.

The hom carriage of the MBOAT7 variant allele was associated with higher rates of non-response (26.7%) than those found in het and wt MBOAT7 patients. The MBOAT7 risk allele has been linked to an increased risk of NAFLD, NASH, alcoholic cirrhosis, and the development of fibrosis in hepatitis B and C [22,38,39,47]. In contrast, only one study could find a beneficial effect of the MBOAT7 variant allele carriage on transplant-free survival in primary sclerosing cholangitis [48]. Considering AIH, an increased risk of HCC has been found to be associated with MBOAT7 [20]. The association of the MBOAT7 homozygosity with non-response in AIH needs to be further evaluated in other AIH cohorts with consideration of the new IAIHG treatment response criteria.

Concerning HSD17B13 and TM6SF2, the lack of an impact on the AIH course and ALT levels in our cohort confirms the available data from Germany and Poland, where no influence of both variants on clinical endpoints, laboratory values, or non-invasive fibrosis scores were observed [19,20].

Our study is the first to describe a combined influence of different SNPs on treatment response in AIH. We calculated a polygenic risk score for each patient based on the individual allele frequencies of PNPLA3, TM6SF2, HSD17B13, and MBOAT7, which correlated with the treatment response. Higher scores were associated with worse treatment responses in a linear regression model. The usefulness of polygenic risk scores has been shown repeatedly for NAFLD cohorts and in multiple other common diseases [49,50,51,52,53,54,55]. However, no such examinations have been performed in AIH patients. The only SNP with a proven effect on outcome in AIH has been PNPLA3 thus far, but evidence for the possible effects of MBOAT7, HSD17B13, and TM6SF2 is lacking. We noticed a higher rate of non-response in hom MBOAT7 risk allele carriers. This observation, combined with the worse long-term biochemical responses in PNPLA3 risk variant carriers, suggest that the value of genetic testing in AIH may lie in the sum of detected variants. We therefore suggest that genetic risk profiling with multiple SNPs for each AIH patient should be evaluated in larger and prospective studies.

Clinical endpoints turned out to be non-significant for all SNP subgroups; however, this might be related to two factors: first, small group sizes, and second, referral bias. The German AIH cohort of Mederacke et al. was considerably larger than our cohort (239 vs. 81 FU patients). In contrast to Hannover, our clinic is not a transplant center. Most patients were directly referred by general practitioners, thus we are covering, in the region, the whole clinical spectrum from the uncomplicated to the severe courses of AIH. The heterogeneity of outcomes in AIH has also been discussed by Janik et al. where no endpoints could be assigned to specific SNPs in the Polish cohort, although these data were collected at a liver transplant center. Additionally, the underlying cause for difference in the observed endpoints may in fact lie in the different treatment responses and not in the impact of variants on the natural course of the disease. Hence, our study adds important data to the field of genetic testing in a rare disease such as AIH.

We have to acknowledge several limitations of our study. First, the retrospective study design made us depend on available documentation and the already taken blood samples for the genotyping of previously diagnosed AIH, which was not achievable in 27/113. Second, we had to rely solely on transaminases for the evaluation of treatment responses, as IgG was not routinely monitored in many patients whose treatment initiation was as long as 25 years ago. As our clinic is no transplant center, we are able to cover a broader spectrum of patients, from light to severe courses of AIH, compared to other studies.

Briefly, comparing our results to the cohorts of Hannover and Warsaw, we could confirm a detrimental effect of the PNPLA3 variants on the AIH course, but not in the endpoints in our cohort. As novel findings, we noted a link in the homozygous MBOAT7 carriage to a worse treatment response, a correlation of treatment response groups to polygenic risk scores, and the male gender as a risk factor for non-response.

In conclusion, our data suggest that several genetic risk variants may impact treatment responses in AIH patients. Further, MBOAT7 wt might be linked to lower histological fibrosis scores, and MBOAT7 homozygosity to higher rates of non-response. The PNPLA3 risk variant may predispose subjects with AIH to a slower response to therapy, following a gene dosage effect. Those that should be closely monitored are particularly males carrying any PNPLA3 risk allele. This is because they inherit two factors for a potentially worse outcome. Our findings provide new evidence that genetic risk assessment is useful in AIH patients, as worse treatment response appears to be driven by the combined effect of multiple SNPs, rather than due to a single SNP.

## Figures and Tables

**Figure 1 jpm-13-00540-f001:**
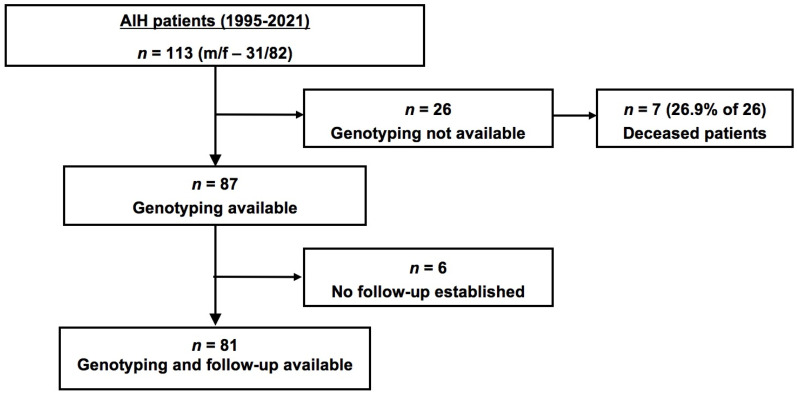
Flow chart of the study cohort. A total of 113 patients were diagnosed with AIH, 87 of those were available for genotyping, and 81 had genotyping and follow-up data available. Abbreviations: AIH, autoimmune hepatitis; m, male; and f, female.

**Figure 2 jpm-13-00540-f002:**
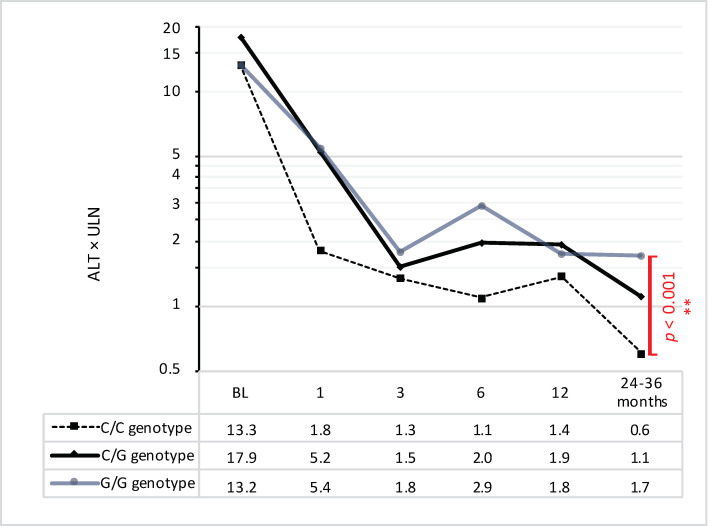
ALT course of the PNPLA3-rs738409 wildtype vs. heterozygous vs. homozygous allele carriers. Logarithmic scale (log_10_) of ALT mean values multiplied by ULN. Levels of significance: ** *p* < 0.001 (Kruskal–Wallis H test). Abbreviations: ALT, alanine aminotransferase; ULN, upper limit of normal.

**Table 1 jpm-13-00540-t001:** Patient characteristics of the whole cohort at baseline.

	Median (IQR) or *n* (%)
Sex (m/f)	31/82 (27.4/72.6%)
Age (years)	57.9 (46.6–67.8)
Cirrhosis at BL (*n*)	14/113 (12.4%)
BMI (kg/m^2^)	25.0 (22.0–27.5)
Cholesterol (mg/dL)	190.8 (147.5–227.0)
CRP (mg/dl)	0.4 (0.1–1.1)
ALT (×ULN)	7.6 (1.9–19.3)
AST (×ULN)	5.9 (1.7–16.4)
ALP (×ULN)	1.2 (0.8–1.8)
GGT (×ULN)	4.3 (1.8–7.1)
Bilirubin (×ULN)	1.1 (0.6–3.1)
Prothrombin time (%)	87.5 (71.2–100.0)
Albumin (g/dL)	1.2 (0.6–8.8)
Hemoglobin (g/dL)	13.9 (13.0–14.7)
WBC (G/L)	5.6 (4.7–7.2)
Thrombocytes (G/L)	201 (159–244)
IgG (g/L)	18.8 (14.1–27.3)
Diabetes mellitus (*n*)	9/113 (8.0%)

Abbreviations: BL, baseline; BMI, body-mass index; CRP, C-reactive protein; ALT, alanine aminotransferase; AST, aspartate aminotransferase; ALP, alkaline phosphatase; GGT, gamma-glutamyl transpeptidase; WBC, white blood cell count; and IgG, immunoglobulin G.

**Table 2 jpm-13-00540-t002:** Clinical endpoints according to different single-nucleotide polymorphisms.

	Wildtype	Heterozygous	Homozygous	*p*-Value
PNPLA3				
Mortality	2/40	1/31	2/10	0.145
ESLD	7/40	7/31	2/10	0.867
TM6SF2				
Mortality	5/67	0/14	0	0.291
ESLD	15/67	1/14	0	0.193
HSD17B13				
Mortality	2/41	3/29	0/11	0.424
ESLD	8/41	7/29	1/11	0.565
MBOAT7				
Mortality	2/25	2/38	1/18	0.900
ESLD	6/25	7/38	3/18	0.939

Abbreviations: ESLD, end-stage liver disease; PNPLA3, patatin-like phospholipase domain-containing protein 3; MBOAT7, membrane-bound O-acyltransferase domain containing 7; TM6SF2, transmembrane 6 superfamily member 2; and HSD17B13, 17-beta-hydroxysteroid dehydrogenase 13.

**Table 3 jpm-13-00540-t003:** Distribution of polygenic risk scores (PRS) and mean PRS per treatment response group.

Polygenic Risk Score	*n* (%)
−2	3 (3.7%)
−1	6 (7.4%)
0	17 (21.0%)
1	26 (32.1%)
2	16 (19.8%)
3	10 (12.3%)
4	3 (3.7%)
PRS (mean ± SD)	Treatment response group (*n*)
0.81 ± 1.42	complete remission (32)
1.03 ± 1.22	partial remission (35)
2.14 ± 1.77	non-response (7)

Abbreviations: PRS, polygenic risk score; PNPLA3, patatin-like phospholipase domain-containing protein 3; MBOAT7, membrane-bound O-acyltransferase domain containing 7; TM6SF2, transmembrane 6 superfamily member 2; and HSD17B13, 17-beta-hydroxysteroid dehydrogenase 13. PRS formula = PNPLA3 (C/C 0, C/G 1, G/G 2) plus TM6SF2 (E/E 0, E/K 1) plus MBOAT7 (G/G 0, C/G 1, C/C 2) minus HSD17B13 (A/A 0, A/dupA 1, dupA/dupA 2).

**Table 4 jpm-13-00540-t004:** Factors associated with non-response.

	OR	95% CI	*p*-Value
Age	0.999	0.951–1.050	0.979
Male gender	14.455	1.806–115.672	0.012 *
PNPLA3-rs738409 mutant allele	0.652	0.096–4.416	0.552
TM6SF2- rs58542926 K-allele	3.340	0.421–26.492	0.254
HSD17B13-rs72613567 dupA	0.245	0.035–1.708	0.156

Abbreviations: CI, confidence interval; OR, odds ratio; PNPLA3, patatin-like phospholipase domain-containing protein 3; TM6SF2, transmembrane 6 superfamily member 2; and HSD17B13, 17-beta-hydroxysteroid dehydrogenase 13. Levels of significance: * *p* < 0.05 (binomial logistic regression).

## Data Availability

The datasets generated and analyzed during the current study are not publicly available due to possible harm of individual privacy, but are instead available from the corresponding authors on reasonable request.

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
