# Peer review of "Genetic Variants Determine Treatment Response in Autoimmune Hepatitis"

_jpm, 2023, doi:10.3390/jpm13030540_

Round 1
Reviewer 1 Report
In this study, the authors investigated the role of single-nucleotide polymorphisms (SNPs) in activity, stage and treatment response of autoimmune hepatitis (AIH) patients. Overall, 113 AIH patients were studied and in 81 genotyping of PNPLA3-rs738409, MBOAT7-rs626238, TM6SF2-rs58542926 and HSD17B13-rs72613567:TA, biochemical and clinical data at baseline and follow-up were available.
After a median time of follow-up was 2.8 years, five patients died, one underwent liver transplantation. PNPLA3-G/G homozygosity was linked to worse treatment response compared to wildtype (ALT 1.7 vs 0.6 x ULN, p<0.001). MBOAT7-C/C homozygosity was linked to non-response vs wt and heterozygosity (p=0.022). Male gender was associated with non-response (OR 14.5, p=0.012) and higher prevalence of PNPLA3 (G/G vs C/G vs wt 41.9/40.0/15.0% males, p=0.03). MBOAT7 wt was linked to less histological fibrosis (p=0.008).
A polygenic risk score based on the individual genetic risk profile and comprising all SNPs correlated with treatment response (p=0.04).
They concluded that genetic risk variants impact treatment response in AIH in a gene-dosage dependent manner. MBOAT7 and PNPLA3 mediated most of the observed effect, explaining in part the predisposition of male subjects to worse treatment response.
The study is of interest however, some issues deserve further details and should be addressed.
-Patients: the authors reported that "We identified 113 cases of AIH between 1996 and 2021....In the majority of patients (97/113, 85.8%), diagnosis of AIH was established at or shortly after first referral in accordance to the simplified diagnostic criteria of the International Autoimmune Hepatitis Group (IAIHG) including autoantibody positivity, elevated IgG, compatible liver histology and absence of viral hepatitis. Liver histology compatible with AIH was available in 97/113 (85.8%) of patients".
However, the simplified AIH score was developed in 2008 and it is likely that in patients diagnosed between 1996 and 2008 the diagnosis was made according to the original revised score (1999), a more comprehensive diagnostic score as previously described (Diagnosis and therapy of autoimmune hepatitis. Mini Rev Med Chem. 2009 Jun;9(7):847-60. ). It would be also important to report the AIH score (definite or probable diagnosis?).
-It is well-known that AIH has a well-defined geneic risk factors. However, geographical differences have been demonstrated between european and north America AIH patients, as previously reported (Evidence of a genetic basis for the different geographic occurrences of liver/kidney microsomal antibody type 1 in hepatitis C. Dig Dis Sci. 2007 Jan;52(1):179-84; Genetic distinctions between autoimmune hepatitis in Italy and North America. World J Gastroenterol. 2005 Mar 28;11(12):1862-6.), and this should be discussed as one of the most important differences in AIH epidemiology and autoantibody profile world-wide.
-Patient Age: according to Table 1, the age of AIH patients was 57.9 (46.6-67.8). However, diagnosis of AIH has been reported in all decades, and nearly 20-30% of diagnosis has been described in elderly patients, who, of interest, display a different genetic background as previously demonstrated (Clinical features of type 1 autoimmune hepatitis in elderly Italian patients. Aliment Pharmacol Ther. 2005 May 15;21(10):1273-7. ).
Author Response
Dear Editors, dear reviewers,
Thank you for your valuable comments and the opportunity to provide a revised version of our manuscript. Adaptations to the manuscript have been made according to your suggestions. Please find a detailed point-by-point response below:
-Patients: the authors reported that "We identified 113 cases of AIH between 1996 and 2021....In the majority of patients (97/113, 85.8%), diagnosis of AIH was established at or shortly after first referral in accordance to the simplified diagnostic criteria of the International Autoimmune Hepatitis Group (IAIHG) including autoantibody positivity, elevated IgG, compatible liver histology and absence of viral hepatitis. Liver histology compatible with AIH was available in 97/113 (85.8%) of patients".
However, the simplified AIH score was developed in 2008 and it is likely that in patients diagnosed between 1996 and 2008 the diagnosis was made according to the original revised score (1999), a more comprehensive diagnostic score as previously described (Diagnosis and therapy of autoimmune hepatitis. Mini Rev Med Chem. 2009 Jun;9(7):847-60. ). It would be also important to report the AIH score (definite or probable diagnosis?).
R: Thank you for the valid comment. Upon reviewing our data, we identified 33 of 113 patients diagnosed with AIH before 2008. Of these patients, 29/33 (88%) had a liver biopsy showing histological features typical for AIH and 31/33 (94%) patients were treated with immunosuppression. Unfortunately, we cannot provide AIH scores, as we are dependent of quality of available documentation and some autoantibodies like LKM, SLA have not been tested especially for diagnoses established many years ago. Those patients where “probable AIH” was determined by the IAIHG scoring systems were re-evaluated according to their response to treatment and on basis of the latter response, diagnosis of “definite AIH” was made. On grounds of these data we are confident that subjects diagnosed before 2008 included in our cohort reliably had AIH.
The following sentence has been added to the manuscript: For patients being diagnosed before 2008, the revised original scoring system for AIH from 1999 was applied and subjects included in the analysis were diagnosed as AIH on the basis of liver biopsy and treatment response.
-It is well-known that AIH has a well-defined geneic risk factors. However, geographical differences have been demonstrated between european and north America AIH patients, as previously reported (Evidence of a genetic basis for the different geographic occurrences of liver/kidney microsomal antibody type 1 in hepatitis C. Dig Dis Sci. 2007 Jan;52(1):179-84; Genetic distinctions between autoimmune hepatitis in Italy and North America. World J Gastroenterol. 2005 Mar 28;11(12):1862-6.), and this should be discussed as one of the most important differences in AIH epidemiology and autoantibody profile world-wide.
R: Thank you for this important note. We have considered this aspect now in the revised version of the manuscript. The following paragraph has been added to the introduction: A role for genetic predisposition in AIH has been suggested previously. In a comparison of Italian and North American cohorts, the HLA-B8-DR3-DQ2 and HLA-DR4 haplotypes occurred more frequently in Northern American AIH patients than in Italian counterparts. Regarding other haplotypes, Italian patients with hepatitis C were more likely to show HLA-DR7 predisposing them to formation of liver kidney microsome (LKM) antibodies.
-Patient Age: according to Table 1, the age of AIH patients was 57.9 (46.6-67.8). However, diagnosis of AIH has been reported in all decades, and nearly 20-30% of diagnosis has been described in elderly patients, who, of interest, display a different genetic background as previously demonstrated (Clinical features of type 1 autoimmune hepatitis in elderly Italian patients. Aliment Pharmacol Ther. 2005 May 15;21(10):1273-7. ).
R: Thank you for the comment. Two paragraphs have been added to the results and discussion sections. Results: To evaluate a possible effect of age on the occurrence of variant alleles and treatment response, we compared younger (< 65 years) and elderly (≥ 65 years) patients similar to Granito et al. Our cohort comprised a similar number of elderly patients ≥ 65 years (36/114, 32%) compared to the mentioned Italian AIH cohort (20/76, 26%). Distribution of HSD17B13, TM6SF2 and MBOAT7 variant alleles between younger and elderly patients were not different. Only the heterozygous PNPLA3 variant was more frequent among elderly vs younger patients (14/23 [60.9%] vs 17/58 [29.3%], p=0.008). Elderly were similar to younger AIH patients regarding treatment responses. Discussion: Comparison of age-related frequencies of SNPs between younger and elderly AIH patients did not yield substantial differences except for PNPLA3 heterozygosity, which was found more often in elderly patients ≥ 65 years. However, this minor difference did not have an impact on treatment response.

Reviewer 2 Report
Dear Authors
Only two minor points are suggested in your manuscript as follows:
1. In the method section, "laboratory evaluation and SNP genotyping" one part or headind is elusive: Missing SNPs were determind from stored DNA samples! Please clarify it.
2. Table 3 shows a polygenic risk score assessment but, there is no explanation for this table inside the discussion. Please discuss it as a part of the findings.
Sincerely
Author Response
Dear Editors, dear reviewers,
Thank you for your valuable comments and the opportunity to provide a revised version of our manuscript. Adaptations to the manuscript have been made according to your suggestions. Please find a detailed point-by-point response below:
1. In the method section, "laboratory evaluation and SNP genotyping" one part or headind is elusive: Missing SNPs were determind from stored DNA samples! Please clarify it.
R: Thank you for this question. We have added the following information in the method section: In our outpatient department, we routinely collect serum specimens of new patients and store them at -20°C for future genetic analysis. Written consent for this procedure is obtained from every patient.
2. Table 3 shows a polygenic risk score assessment but, there is no explanation for this table inside the discussion. Please discuss it as a part of the findings.
R: Thank you for the comment. We have inserted the following explanation in the discussion section: We calculated a polygenic risk score for each patient based on the individual allele frequencies of PNPLA3, TM6SF2, HSD17B13, and MBOAT7, which correlated with treatment response. Higher scores were associated with worse treatment response in a linear regression model.

Round 2
Reviewer 1 Report
The revised manuscript has prperly addressed the raised points and the manuscript cna be accepted.